# Persistent Sleep Quality Deterioration among Post-COVID-19 Patients: Results from a 6-Month Follow-Up Study

**DOI:** 10.3390/jpm12111909

**Published:** 2022-11-16

**Authors:** Evgenia Kalamara, Athanasia Pataka, Afroditi Boutou, Evangelia Panagiotidou, Athina Georgopoulou, Evangelos Ballas, Diamantis Chloros, Symeon Metallidis, Ioannis Kioumis, Georgia Pitsiou

**Affiliations:** 1Respiratory Failure Unit, General Hospital of Thessaloniki “G. Papanikolaou”, 57010 Thessaloniki, Greece; 2Respiratory Medicine Department, Hippokration Hospital, 54642 Τhessaloniki, Greece; 3Department of Respiratory Medicine, General Hospital of Thessaloniki “G. Papanikolaou”, 57010 Thessaloniki, Greece; 4Department of Internal Medicine, General Hospital of Chalkida, 34100 Chalkida, Greece; 5First Department of Internal Medicine, AHEPA University General Hospital, 54636 Thessaloniki, Greece

**Keywords:** post-COVID-19, sleep quality, insomnia, Pittsburgh sleep quality index, SARS-CoV-2

## Abstract

Background: To date, evidence about sleep disturbances among post-COVID-19 patients is limited. This study aimed to evaluate sleep quality after hospitalization due to SARS-CoV-2 infection. Methods: In-person follow-up was conducted in patients with prior hospitalization due to COVID-19 1(Τ1), 3(Τ2), and 6 (Τ3) months after hospital discharge. Patients were asked to complete questionnaires concerning sleep quality: the Pittsburgh Sleep Quality Index (PSQI), the Epworth Sleepiness Scale (ESS), the Athens Insomnia Scale (AIS), the Fatigue Severity Scale (FSS), and the Stop-BANG (S-B) questionnaire. Results: In total, 133 patients were enrolled (mean age: 56.0 ± 11.48 years, 59.4% males). The most frequently reported comorbidity was arterial hypertension (29.8% of patients), while 37.4% of patients had no comorbidities. The majority of participants exhibited poor sleep quality (global PSQI ≥ 5) at T1 (84.3%), T2 (75.7%), and T3 (77.4%). Insomnia was observed in 56.5%, 53.5%, and 39.2% of participants, respectively (AIS ≥ 6). An FSS score ≥ 4 was observed in 51.2%, 33.7%, and 29.1% of participants at T1, T2, T3, respectively. Elapsed time was found to be negatively and independently associated with the global PSQI, PSQI C5-Sleep disturbance, PSQI C7-Daytime dysfunctions, FSS, and AIS after adjustment for possible confounders. No significant difference was found between groups with good and poor sleep quality (based on the global PSQI) with respect to gender (*p* = 0.110), age (*p* = 0.528), BMI (*p* = 0.816), smoking status (*p* = 0.489), hypertension (*p* = 0.427), severity of disease (*p* = 0.224), the Charlson Comorbidity Index (*p* = 0.827), or the length of hospital stay (*p* = 0.162). Participants with excessive daytime sleepiness (EDS) and patients with severe fatigue (FSS ≥ 4) were significantly younger. Females presented a higher rate of insomnia symptoms (55.7% vs. 44.3%, *p* < 0.001). Conclusions: Several sleep disturbances were observed after hospital discharge for COVID-19 pneumonia at certain time points; However, the improvement over time was remarkable in most domains of the assessed questionnaires.

## 1. Introduction

As of July 2022, more than 500 million people have been infected with SARS-CoV-2 worldwide, while more than 6,000,000 deaths have been recorded due to COVID-19 [1]. Unfortunately, apart from the possible detrimental effects of acute infection, a noteworthy proportion of patients develop long-term complications. Signs and symptoms of COVID-19 that are present from 4 weeks up to 12 weeks are defined as ongoing symptomatic COVID-19, and when signs and symptoms persist for more than 12 weeks, post-COVID-19 syndrome is diagnosed. These long-term sequelae may involve not only the respiratory system but almost any organ. Neurological/psychiatric symptoms such as cognitive impairment, headache, peripheral neuropathy, dizziness, delirium, depression, and anxiety, as well as sleep disturbance, may be reported in the clinical presentation of the post-COVID-19 syndrome [2].

During the SARS-CoV-2 outbreak, sleep disorders and impaired sleep quality were studied in various populations, mainly with the aid of questionnaires. COVID-19-related sleep disorders are estimated to affect 30-35% of the general population and healthcare workers, and the frequency is even higher in patients with acute COVID-19 [3,4,5]. Regarding sleep quality, there are some studies that have reported insomnia specifically as a symptom of the post-COVID-19 syndrome. A Chinese follow-up study found that 26% of patients complained about sleep deterioration after the acute infection [6]. Another study comparing sleep disturbances amongst non-COVID-19, COVID-19-positive, and post-COVID-19 patients using the Insomnia Severity Index revealed that patients with post-COVID-19 syndrome presented a significantly higher incidence of insomnia, with a concomitant deterioration of life quality [7]. Among COVID-19 survivors, 28% were found to suffer from post-traumatic stress (PTSD), 31% were found to suffer from depressive disorder, 42% were found to suffer from an anxiety disorder, and 40% were found to suffer from insomnia [8].

However, so far, few studies have been conducted specifically addressing post-COVID-19 sleep impairment. Isolation, hospitalization, and stress are important factors [9], but the exact mechanism behind COVID-19-related sleep disturbances, as well as the duration of such manifestations, remain to be investigated. In our study, we tried to thoroughly evaluate sleep quality using multiple structured questionnaires and in-person visits, allowing clinical assessment and sequential follow-ups in order to evaluate potential changes in clinical features over time.

## 2. Materials and Methods

### 2.1. Participant Selection

This study was conducted at the General Hospital of Thessaloniki “G. Papanikolaou”, Greece, from August 2020 to February 2022 with patients referred to the post-COVID-19 outpatient clinic. Patients over 18 years of age with prior hospitalization due to COVID-19 and laboratory-confirmed SARS-CoV-2 infection who consented to participate were included.

### 2.2. Data Collection

Patients were followed-up in person 1(Τ1), 3(Τ2), and 6(Τ3) months after hospital discharge. Details about their medical history were recorded, and each visit included a clinical examination as well as common baseline investigations (chest x-ray and routine blood testing). Patients were divided into four groups, according to the severity of the disease: group 1—mild disease with no need for oxygen therapy during hospitalization, group 2—moderate disease with the need for oxygen therapy (OT) during hospitalization, group 3—severe disease with the need for noninvasive ventilation (NIV) or a high-flow nasal cannula (HFNC), and group 4—very severe disease with the need for invasive ventilation (IV) with or without the use of extracorporeal membrane oxygenation (ECMO).

Sleep quality was estimated with the aid of the following questionnaires: the Pittsburgh Sleep Quality Index (PSQI), the Epworth Sleepiness Scale (ESS), the Athens Insomnia Scale (AIS), the Fatigue Severity Scale (FSS), and the Stop-BANG questionnaire (S-B). Interviews were conducted by the same team of respiratory physicians. The PSQI is a self-reported questionnaire. It estimates sleep quality over a period of one month. It is comprised of seven different components: subjective sleep quality, sleep latency, sleep duration, habitual sleep efficiency, sleep disturbances, the use of sleeping medication, and daytime dysfunction. The component scores are added up to produce a global score [10]. A global PSQI score ≥ 5 was used to distinguish poor sleepers from good sleepers. The ESS is another self-rated questionnaire. Interviewees rate their tendency to fall asleep in eight distinct situations from 0 to 3. A high ESS is related to excessive somnolence [11]. A score > 10 on the ESS was used to determine excessive daytime sleepiness (EDS). Specifically, an ESS score of 0–10 was considered normal, 11–14 indicated mild sleepiness, 15–17 indicated moderate sleepiness, and 18–24 indicated severe sleepiness. The AIS is a self-administered questionnaire that consists of five components concerning sleep induction, night-time awakenings, the final awakening, and the duration and quality of sleep, and three components regarding overall well-being, functional ability, and daytime somnolence [12]. Insomnia was characterized as mild, moderate, or severe with AIS scores of 6–9, 10–15, and 16–24, respectively, while an AIS score < 6 was considered normal (no insomnia). The FSS is a nine-item self-rated scale about fatigue, its severity, and how it affects certain activities. Answers are scored on a seven-point scale where 1 = strongly disagree and 7 = strongly agree. The higher the score, the more severe the fatigue and the more it affects the person’s activities [13]. Severe fatigue was determined by an FSS score ≥ 4. The Stop-BANG questionnaire includes eight yes or no questions based on the clinical characteristics of sleep apnea. A responder’s OSA risk is estimated based on the total score, which has a range from 0 to 8. [14]. The risk of OSA was categorized as low, intermediate, or high with S-B scores of ≤2, 3–4, and ≥5, respectively.

### 2.3. Statistical Analysis

Comparisons of the questionnaire scores between the three time points (T1, T2, and T3) were conducted using a one-way repeated-measures analysis of variance (ANOVA) with “Time” as a within-subject factor. Post hoc comparisons were performed using the least significant difference (LSD) method. Partial eta squared (η_p_^2^) was used as a measure of effect size.

A linear mixed-model regression (LMM) was further used to assess differences in the main outcome measures. Mixed models were estimated separately for each outcome variable, with time, sex, age, BMI, severity, and the Charlson Comorbidity Index entered as fixed factors. For each linear mixed model regression, a random intercept was set regarding the repeated measures of each participant.

The statistical analysis was performed with IBM SPSS Statistics version 25.0. Values of *p* ≤ 0.05 were considered statistically significant.

## 3. Results

A total of 133 subjects were included. The clinical characteristics of the participants are shown in Table 1. The majority were males (59.4%) with a mean (SD) age of 56.0 (11.48) years. Their median BMI was 29, which characterized them as overweight. Most patients (60.6%) had moderate severity of disease (group 2). The mean duration of hospitalization was approximately 13 days. Based on the Charlson Comorbidity Index, the severity of comorbidity was moderate, as the median value of the sample was 2. The most frequent comorbidity was arterial hypertension (29.8% of patients), while 37.4% of patients reported no comorbidity.

Using a multiple regression analysis, no significant difference was found between patients with good vs. poor sleep quality (PSQI ≥ 5) with respect to gender (*p* = 0.110), age (*p* = 0.528), BMI (*p* = 0.816), smoking status (*p* = 0.489), hypertension, (*p* = 0.427), the severity of disease (*p* = 0.224), the Charlson Comorbidity Index (*p* = 0.827), or the length of hospital stay (*p* = 0.162).

Participants with EDS were significantly younger than participants without EDS (median: 48 vs. 57 years, *p* = 0.002). Moreover, patients without EDS presented more comorbidities than patients with EDS (median: 2 vs. 0, *p* = 0.001).

Participants with severe fatigue (FSS ≥ 4) were significantly younger (mean: 53 vs. 59.1, *p* = 0.004), more obese (mean BMI: 30.5 vs. 28.6, *p* = 0.043), and had a lower Charlson Comorbidity Index (median: 1 vs. 2, *p* = 0.042) than participants without severe fatigue.

A statistically significant difference was noticed between genders on the AIS (*p* < 0.001), with females presenting a higher rate of insomnia symptoms (55.7% vs. 44.3%, *p* < 0.001). Furthermore, participants with insomnia had significantly higher BMI values (mean: 30.6 vs. 28.5, *p* = 0.018) and fewer hospital days (median: 8 vs. 12, *p* = 0.014). A statistically significant difference was also found between genders in terms of the severity of insomnia (*p* < 0.001), with more severe (60% vs. 40%) insomnia symptoms being observed mostly in females (see Appendix A).

A one-way repeated-measures ANOVA was conducted to compare the sleep patterns across the three different time points. Although the results did not show a significant main effect of “Time” for the measured variables, some of the comparisons between the time points reached significance. These comparisons concerned the following variables: PSQI—Global score (F2,146 = 2.840, *p* = 0.062, ηp^2^ = 0.037), PSQI C2—Sleep latency (F2,146 = 3.026, *p* = 0.060, ηp^2^ = 0.040), PSQI C3—Sleep duration (F2,146 = 2.485, *p* = 0.087, ηp^2^ = 0.033), PSQI C5—Sleep disturbance (F2,146 = 2.581, *p* = 0.079, ηp^2^ = 0.034), PSQI C7—Daytime dysfunctions (F2,146 = 3.571, *p* = 0.031, ηp^2^ = 0.047), the Fatigue Severity Scale (F2,142 = 8.026, *p* = 0.001, ηp^2^ = 0.102), and the Athens Insomnia scale (F2,140 = 2.997, *p* = 0.065, ηp^2^ = 0.041). The results from significant LSD post hoc comparisons are reported as follows: The mean PSQI—Global scores at T1, T2, and T3 were 8.11 (SD = 3.87), 7.22 (SD = 3.63), and 7.53 (SD = 3.72), respectively; the difference between T1 and T2 was statistically significant (*p*_1–2_ = 0.021). Regarding PSQI C2—Sleep latency, the difference between T1 and T2 was statistically significant (*p*_1–2_ = 0.023). Concerning PSQI C3—Sleep duration, the difference between T2 and T3 was statistically significant (*p*_2–3_ = 0.021). For PSQI C5—Sleep disturbance, the difference between T1 and T3 was statistically significant (*p*_1–3_ = 0.047). For PSQI C7—Daytime dysfunctions, the difference between T1 and T3 was statistically significant (*p*_1–3_ = 0.011). In particular, the LSD post hoc comparisons revealed significant reductions in the PSQI—Global score and the PSQI C2—Sleep latency score at T2 compared to T1. Moreover, a significant reduction was also observed in the PSQI C3—Sleep duration score at T2 compared to T3. Finally, we also observed that, compared with the T1, there were significant reductions in the PSQI C5—Sleep disturbance score and the PSQI C7—Daytime dysfunctions score at T3.

The mean FSS scores at T1, T2, and T3 were 3.41 (SD = 1.73), 2.93 (SD = 1.62), and 2.71 (SD = 1.48), respectively. The difference between T1 and T2 was statistically significant (*p*_1–2_ = 0.011), as was the difference between T1 and T3 (*p*_1–3_ = 0.001). The mean Athens Insomnia Scale scores at T1, T2, and T3 were 6.24 (SD = 4.75), 5.63 (SD = 4.40), and 5.14 (SD = 5.00), respectively; the difference between T1 and T3 was statistically significant (*p*_1–3_ = 0.042). A significant reduction in the FSS score was observed at T3 compared to the FSS scores at T1 and T2. Moreover, the Athens Insomnia Scale was decreased at T3 compared to T1 (Table 2 and Table 3; see Appendix A).

A linear mixed-model regression analysis was additionally conducted to identify whether time had an independent effect on the various sleep disturbance scales after adjusting for age, BMI, gender, disease severity, and the Charlson Comorbidity Index. The main effect of time was significant on the FSS Scale (b = −0.376, *p* < 0.001), AIS (b = −0.487, *p* = 0.046), PSQI—Global score (b = −0.427, *p* = 0.010), PSQI C5—Sleep disturbance (b = −0.122, *p* = 0.001), and PSQI C7—Day Time dysfunctions (b = −0.16, *p* < 0.001), with decreasing scores over time. Time had no effect on the ESS, the STOP-Bang scale, or the remaining PSQI components (see Appendix A).

## 4. Discussion

The burden of sleep deterioration among post-COVID-19 patients was specifically addressed and assessed at consecutive time points in order to study the development of this clinical manifestation and its long-term outcome. By evaluating overall sleep quality with the aid of the PSQI, our study revealed that a significant percentage of COVID-19 survivors with prior hospitalization report sleep disturbances at various time points during follow-up, in accordance with evidence that 50–75% of COVID-19 patients suffer from sleep deterioration [3,4,5]. However, apart from the PSQI domain of daytime dysfunction and the severity of fatigue as assessed by the FSS, most sleep disturbances tended to improve with time in our study. To the authors’ knowledge, this is the first longitudinal study to indicate that elapsed time is independently and negatively associated with the presence of sleep disturbances during a six-month follow-up after hospitalization for COVID-19 pneumonia. Lin et al. concluded that the occurrence of sleep disturbances in previously hospitalized COVID-19 patients ranges from 29.5 to 40%. [15]. A previous study regarding sleep deterioration after acute SARS-CoV-2 infection reported a high insomnia severity index (13.01 ± 4.9) and PSQI (15.37 ± 4.43) as well as high scores on various items of the scale of quality of life [16]. Yet another longitudinal study that compared the sleep quality of patients before infection and 1 month after discharge also showed that the mean PSQI score was significantly higher in the post-COVID-19 group compared to the pre-COVID-19 group [17]. These findings are in accordance with the results of our study, which revealed high scores on the PSQI (8.11 ± 3.87, 7.22 ± 3.63, and 7.53 ± 3.72 during the first, second, and third follow-ups, respectively) and the AIS (6.24 ± 4.75, 5.63 ± 4.40, and 5.14 ± 5 during the first, second, and third follow-ups, respectively).

Apart from observing the presence of post-COVID-19 sleep disturbances, it was also interesting to investigate the duration of the manifestations and the types of sleep impairment. Our findings indicate that 6 months after discharge, sleep disturbances were significantly improved. During the post-COVID-19 recovery period, significant changes over time were observed for the global PSQI as well as for certain PSQI components: sleep disturbance and daytime dysfunction. Fatigue and insomnia also showed amelioration over time. Notably, in a meta-analysis by Jahrami et al. concerning sleep problems during the COVID-19 pandemic by population, the mean PSQI for all different populations was 7.1 (95% CI: 6.3–8.0), while the score for the general population was 6.0 (95% CI: 5.3–6.8). The highest scores were found for the PSQI components sleep latency, sleep disturbances, and sleep duration [5].

Concerning patient characteristics and their association with poor sleep, a 2-month follow-up study in which COVID-19 patients were screened for sleep disturbances during hospitalization concluded that 35% of participants were still suffering from poor sleep quality. Interestingly, age and diabetes mellitus were independently associated with sleep disturbances, while diabetic patients were found to be 65% more likely to report sleep disturbances [18]. Another cohort study found that the burden of sequela was higher in people with poorer baseline health status and increased according to the severity of the acute infection. Moreover, certain post-COVID-19 manifestations, such as sleep disturbances, smell disorders, affective disorders, and headaches, were more evident in younger patients. [19].

In our study, age, smoking status, BMI, gender, arterial hypertension (which was the most frequent comorbidity), the severity of disease, the Charlson Comorbidity Index, and the length of hospital stay were not related to the overall sleep quality or its change over time. However, participants with excessive daytime sleepiness and participants with severe fatigue were significantly younger [20]. As mentioned above, this finding is in accordance with the study by Xie et al., in which sleep disorders were more prominent in younger adults [19]. Although sleep disorders are usually more frequent in older patients [21], different factors may contribute to these findings. An interesting study by Zitting et al. reported that young adults are more susceptible to chronic sleep deprivation and recurrent circadian interruption than their older counterparts. The protocol included sleep reduction with circadian disruption over three weeks. Younger and older participants described themselves as equally somnolent—However, the younger adults presented with more neurobehavioral deficiencies as a result of sleepiness [22]. In our study, sleep disturbances were evaluated with questionnaires. Whether sleep deficiency was indeed more prominent in younger patients or young patients were more affected by those same sleep deficiencies remains to be clarified by further studies. As far as the gender differences are concerned, in our study, women seemed to be more afflicted by insomnia than men, presenting with a higher rate of symptoms and increased severity of insomnia. This finding is consistent with previous research that indicated that women are more prone to insomnia and mental health problems [20]. Notably, a review by Krishnan et al. concluded that although women have longer sleep durations, shorter sleep-onset latencies, and better sleep efficiency, they report more sleep-related issues than men, while insomnia-relevant studies suggest the predominance of women. Additionally, menstruation, pregnancy, and menopause can affect sleep characteristics [23].

Finally, apart from the patients’ traits, in our study, the linear mixed-model regression indicated the significance of time as an independent factor affecting the clinical course of sleep disturbance after adjusting for age, BMI, gender, disease severity, and the Charlson Comorbidity Index. We conclude that time seemed to be the most important parameter affecting the natural course of post-COVID-19 sleep disorders in our study. However, it seems that further research is needed to determine whether certain patient characteristics are related to sleep disturbances and if so, to clarify the underlying pathophysiological mechanisms.

Regarding the etiology of poor sleep quality in these patients, several different factors may be implicated. Lockdown, quarantine, fear of the disease, atypical homework schedules, and reduced exposure to daylight can take a toll on the circadian rhythm [24,25,26,27,28]. Psychiatric conditions certainly seem to be a significant burden for COVID-19 survivors, and therefore they could be linked to the observed sleep disturbance. A large retrospective cohort study by Taquet et al. reported significant neurological and psychiatric morbidity in the first 6 months after acute infection. The prevalences of anxiety disorders and psychotic disorders in the COVID-19 cohort were 17.39% (17.04–17.74) and 1.40% (1.30–1.51), respectively [29]. Further research to analyze the relationship of sleep variables with anxiety, depression, and sleep hygiene would be of value. Yet another factor could be the viral infection itself due to the neurotropic potential of SARS-CoV-2 [30]. Notably, at least three types of encephalitis/encephalopathy are involved in COVID-19 (direct viral infection, infection due to systemic inflammation, and an autoimmune response) [31]. A relative meta-analysis concluded that SARS-CoV-2 might cause delirium in the acute stage, while neuropsychiatric syndromes are possible in the longer term [32]. Some authors even assume that COVID-19 might be related to the postinfectious myalgic encephalomyelitis/chronic fatigue syndrome [33]. The presence of viral proteins in the central nervous system, the state of systemic inflammation, and oxygen desaturation may be responsible for the disruption of sleep patterns [8,18,34,35,36,37,38]. Perhaps there is also a link between prior sleep disturbance and susceptibility to the virus, as sleep deprivation may have a negative impact on the immune response [39,40].

To conclude, sleep disturbance and fatigue are often present after hospitalization for COVID-19 pneumonia; however, most but not all of these disturbances tend to improve with time. The data reveal that sleep problems can remain an important burden for post-COVID-19 patients, regardless of the severity of the disease. Moreover, there are indications that younger patients may be more afflicted by fatigue and excessive daytime sleepiness. Further research is needed to clarify the causative mechanisms behind these findings. The ideal instruments and methods to assess sleep disturbances in these patients also remain to be defined. Based on our research, the Epworth Sleepiness Scale and the STOP-Bang Scale may not be optimal screening tools for such evaluation compared to the PSQI, AIS, and FSS.

## 5. Conclusions

The results suggest that persistent and significant sleep disturbances last up to 6 months after hospital discharge, although remarkable amelioration was observed over time.High scores on the PSQI, FSS, and Athens insomnia scales were noted in a significant percentage of patients. After adjustment for possible confounders, these scores were found to decrease over time.No significant differences were found between the groups with good and poor sleep quality (based on the global PSQI) with respect to gender, age, BMI, smoking status, hypertension, the severity of disease, the Charlson Comorbidity Index, or the length of hospital stay.Participants with excessive daytime sleepiness (EDS) and participants with severe fatigue were significantly younger.A statistically significant difference was noticed between males and females on the Athens Insomnia Scale, with females showing a higher rate of insomnia symptoms.

## 6. Limitations

The findings of this study have to be seen in light of some limitations. The evaluation of sleep quality was based on questionnaires. Therefore, by definition, there is an element of subjectivity. However, patient assessment was conducted by the same team of three physicians, minimizing subjectivity errors. Perhaps further research could incorporate polysomnography or other more objective tests in addition to questionnaires. Furthermore, during the course of the pandemic, different variables of SARS-CoV-2 were prominent in the general population, possibly causing differences in disease manifestations. The present study did not distinguish different SARS-CoV-2 variables, mainly due to a lack of available data in the hospitalization records. Finally, the role of psychiatric conditions in sleep quality was not separately analyzed, as the available patients’ medical records were not sufficient to distinguish pre-existing and new psychiatric manifestations.

## Figures and Tables

**Table 1 jpm-12-01909-t001:** Clinical characteristics of the participants.

	N
**Gender**	
Male, n (%)	79(59.4%)
Female, n (%)	54 (40.6%)
**Age**, mean (SD)	56.0 (11.48)
**BMI**, median (IQR)	29 (26–32.7)
**Smoking status**	
Never smoked, n (%)	72 (55%)
Former smoker, n (%)	45 (34.4%)
Current smoker, n (%)	14 (10.7%)
**No comorbidities**	
Yes, n (%)	49 (37.4%)
No, n (%)	82 (62.6%)
**Arterial Hypertension, n (%)**	39 (29.8%)
**Cardiovascular disease, n (%)**	22 (16.8%)
**Chronic obstructive pulmonary disease, n (%)**	15 (11.5%)
**Diabetes, n (%)**	14 (10.7%)
**Autoimmune disease, n (%)**	13 (9.9%)
**Nervous system diseases, n (%)**	5 (3.8%)
**Active neoplasms, n (%)**	5 (3.8%)
**Chronic kidney disease, n (%)**	4 (3.1%)
**Severity**	
Group 1, n (%)	10 (8.2%)
Group 2, n (%)	74 (60.6%)
Group 3, n (%)	24 (19.7%)
Group 4, n (%)	14 (11.5%)
**Hospitalization duration (in days),** mean (SD)	12.58 (8.90)
**Charlson Comorbidity Index,** median (IQR)	2 (1–3)

**Table 2 jpm-12-01909-t002:** Changes in the PSQI components over the three time points.

	1st Timepoint	2nd Timepoint	3rd Timepoint	F-Value	*p*	H_p_^2^
	Mean (SD)	Mean (SD)	Mean (SD)			
PSQI—Global score	8.11 (3.87)	7.22 (3.63)	7.53 (3.72)	2.840	0.062 ^a^	0.037
PSQI C1—Subjective sleep quality	0.99 (0.77)	0.85 (0.73)	0.95 (0.83)	1.219	0.299	0.016
PSQI C2—Sleep latency	1.14 (0.91)	0.88 (0.81)	1.01 (0.85)	3.026	0.060 ^b^	0.040
PSQI C3—Sleep duration	1.27 (1.01)	1.20 (0.95)	1.43 (0.98)	2.485	0.087 ^c^	0.033
PSQI C4—Habitual sleep efficiency	2.26 (1.28)	2.14 (1.36)	2.2 (1.32)	0.230	0.795	0.003
PSQI C5—Sleep disturbance	1.39 (0.59)	1.26 (0.60)	1.22 (0.58)	2.581	0.079 ^d^	0.034
PSQI C6—Use of sleep medication	0.42 (0.96)	0.37 (0.81)	0.37 (0.86)	0.217	0.805	0.003
PSQI C7—Daytime dysfunctions	0.66 (0.73)	0.49 (0.69)	0.41 (0.57)	3.571	**0.031**	0.047

Notes: ^a^. PSQI—Global score: *p*_1–2_ = 0.021, ^b^. PSQI C2—Sleep latency: *p*_1–2_ = 0.023, ^c^. PSQI C3—Sleep duration: *p*_2–3_ = 0.021, ^d^. PSQI C5—Sleep disturbance: *p*_1–3_ = 0.047.

**Table 3 jpm-12-01909-t003:** Changes in the different assessment tools over the three time points.

	1st Timepoint	2nd Timepoint	3rd Timepoint	F-Value	*p*	η_p_^2^
	Mean (SD)	Mean (SD)	Mean (SD)			
Epworth Sleepiness Scale	5.86 (4.08)	5.65 (3.59)	5.47 (3.70)	0.550	0.578	0.008
FSS Scale	3.4 1(1.73)	2.93 (1.62)	2.71 (1.48)	8.026	**0.001 ^e^**	0.102
STOP-Bang	2.61 (1.37)	2.65 (1.30)	2.69 (1.44)	0.232	0.793	0.003
Athens Insomnia Scale	6.24 (4.75)	5.63 (4.40)	5.14 (5.00)	2.997	0.065 ^f^	0.041
		**1st Timepoint**	**2nd Timepoint**	**3rd Timepoint**		
		**N (%)**	**N (%)**	**N (%)**		
**Global PSQI**						
<5		18 (15.7%)	25 (24.3%)	21 (22.6%)		
≥5		97 (84.3%)	78 (75.7%)	72 (77.4%)		
**Epworth sleepiness scale**						
No EDS (≤10)		103 (85.1%)	79 (80.6%)	73 (92.4%)		
EDS (>10)		18 (14.9%)	19 (19.4%)	6 (7.6%)		
**Epworth Sleepiness Scale**						
Normal Sleepiness (0–10)		103 (85.1%)	79 (80.6%)	73 (92.4%)		
Mild sleepiness (11–14)		13 (10.7%)	14 (14.3%)	5 (6.3%)		
Moderate Sleepiness (15–17)		3 (2.5%)	5 (5.1%)	0 (0.0%)		
Severe Sleepiness (18–24)		2 (1.7%)	0 (0.0%)	1 (1.3%)		
**FSS Scale**						
<4		59 (48.8%)	65 (66.3%)	56 (70.9%)		
≥4		62 (51.2%)	33 (33.7%)	23 (29.1%)		
**STOP-Bang**						
Low risk (≤2)		54 (44.6%)	46 (46.9%)	40 (50.6%)		
Intermediate risk (3–4)		51 (42.1%)	37 (37.8%)	32 (40.5%)		
High risk (≥5)		16(13.3%)	15 (15.3%)	7 (8.9%)		
**Athens Insomnia Scale**						
<6		54 (43.5%)	46 (46.5%)	48 (60.8%)		
≥6		70 (56.5%)	53 (53.5%)	31 (39.2%)		
**Athens Insomnia Scale**						
Absence of insomnia (0–5)		54 (43.5%)	46 (46.5%%)	48 (60.8%)		
Mild (6–9)		30 (24.2%)	27 (27.3%)	11 (13.9%)		
Moderate (10–15)		30 (24.2%)	19 (19.2%)	14 (17.7%)		
Severe (16–24)		10 (8.1%)	7 (7.1%)	6 (7.6%)		

Notes: ^e^. FSS: *p*_1–2_ = 0.011, *p*_1–3_ = 0.001. ^f^. AIS: *p*_1–3_ = 0.042.

## Data Availability

The data presented in this study are available on request from the corresponding author. The data are not publicly available due to an institutional ethics policy.

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
