# Peer review of "Persistent Sleep Quality Deterioration among Post-COVID-19 Patients: Results from a 6-Month Follow-Up Study"

_jpm, 2022, doi:10.3390/jpm12111909_

Round 1
Reviewer 1 Report
Dear authors,
thank you for your manuscript. I have some concerns about the results and discussion section and thus also for the abstract.
At the end of the method section, you explicitly set the significance level at p =/> 0.05. Nevertheless, you report results as significant which are above this significance level (I have marked the results in the manuscript) and subsequently interpret these results in the discussion as significant. This has to be rectified.
There is a marked difference between the quality of the introduction and method section and the results section. There are repeatedly some very strange sentences where I suspect that brackets should have been inserted or something similar. I have marked them in the manuscript.
Also, the interpretation of the age and gender difference could be a bit more thorough, especially the age effect.
Best wishes

Author Response
Response to Reviewer 1 Comments
Dear Colleague,
Thank you for your prompt evaluation and detailed report.
Point 1: At the end of the method section, you explicitly set the significance level at p =/> 0.05. Nevertheless, you report results as significant which are above this significance level (I have marked the results in the manuscript) and subsequently interpret these results in the discussion as significant. This has to be rectified.
Response 1: We really appreciate this comment. We totally agree that, in an effort to simplify the presentation of results for the readers, as well as to not make the article overwhelmingly lengthy, we explained certain findings in less detail, which could have created some confusion. The p values that you noted are the total p values, concerning T1 through T3. However, in certain variables, significant differences were found between T1 and T2 , or T2 and T3, etc (by applying post hoc comparisons). Following your comment, we modified the text, tables 2 and 3, as well as the supplementary material accordingly, in order to be more precise.
Point 2: There is a marked difference between the quality of the introduction and method section and the results section. There are repeatedly some very strange sentences where I suspect that brackets should have been inserted or something similar. I have marked them in the manuscript.
Response 2: Thank you for this comment. All marked sentences have been carefully amended, following this remark, to improve paper quality.
Point 3: Also, the interpretation of the age and gender difference could be a bit more thorough, especially the age effect.
Response 3: Following this comment we have added some further information concerning the age and sex differences, as you proposed, with corresponding references
All the additions and changes made have been clearly indicated in the revised version of our manuscript are marked with red color using the”Track Changes” function of MSWord , as requested. Certain further text modifications were additionally made to the original manuscript according to the similarity report that was provided .Please note that there is also an added section in the supplementary material. A clean copy of the revised manuscript
is also submitted.
Should you have any additional remarks or questions, please do not hesitate to contact me.
Kind regards,
Evgenia Kalamara
Reviewer 2 Report
This is a relevant and inovative prospective study on sleep quality in times of COVID-19 where sleep quality indicators were mainly related to physical variables, but not to psychological ones. However, it is known that sleep quality is closely associated with psychological distress and poor sleep hygiene, in addition, Taquet et al, emphasized that there is a weaker relationship between COVID-19 severity markers and psychiatric disorders compared to neurological disorders. This would indicate that the former seems to have more psychological implications than a direct manifestation of COVID-19.
Thus, it would have been important for this study to analyze the relationship of sleep variables with anxiety, depression, sleep hygiene, and consequently, expand the paragraph on the etiology of poor sleep quality. It would also be interesting to include the reference by Taquet et al.
Taquet M, Geddes JR, Husain M, Luciano S, Harrison PJ. 6-month neurological and psychiatric outcomes in 236 379 survivors of COVID-19: a retrospective cohort study using electronic health records. Lancet Psychiatry. 2021;8:416-427.
Author Response
Response to Reviewer 2 Comments
Dear Sir/Madam,
Thank you for your thorough evaluation and enlightening report.
Point 1: This is a relevant and inovative prospective study on sleep quality in times of COVID-19 where sleep quality indicators were mainly related to physical variables, but not to psychological ones. However, it is known that sleep quality is closely associated with psychological distress and poor sleep hygiene, in addition, Taquet et al, emphasized that there is a weaker relationship between COVID-19 severity markers and psychiatric disorders compared to neurological disorders. This would indicate that the former seems to have more psychological implications than a direct manifestation of COVID-19. Thus, it would have been important for this study to analyze the relationship of sleep variables with anxiety, depression, sleep hygiene, and consequently, expand the paragraph on the etiology of poor sleep quality.
Response 1: We really appreciate this comment. It would be, indeed, useful to separately analyze the relationship between psychiatric variables and sleep quality. However, a thorough evaluation of psychiatric manifestations was not within the initial outcomes of the study, while the available patients’ medical records were not sufficient to distinguish pre-existing and new psychiatric manifestations; thus we focused on the variables deriving from the available data. Following this comment, we included this fact in a new section of limitations and we made certain relative changes to the text, as proposed.
Point 2: It would also be interesting to include the reference by Taquet et al.
Taquet M, Geddes JR, Husain M, Luciano S, Harrison PJ. 6-month neurological and psychiatric outcomes in 236 379 survivors of COVID-19: a retrospective cohort study using electronic health records. Lancet Psychiatry. 2021;8:416-427.
Response 2: Following this suggestion, the reference is added in the reference list, and its findings are mentioned in the section of discussion.
All the additions and changes made have been clearly indicated in the revised version of our manuscript are marked with red color using the”Track Changes” function of MSWord , as requested. Certain further text modifications were additionally made to the original manuscript according to the similarity report that was provided .Please note that there is also an added section in the supplementary material. A clean copy of the revised manuscript
is also submitted.
Should you have any additional remarks or questions, please do not hesitate to contact me.
Kind regards,
Evgenia Kalamara
Round 2
Reviewer 1 Report
Dear authors,
thank you very much for your revised version of the manuscript. I think the manuscript is quite improved. Nevertheless, I think it is quite problematic that the non-significant main effects of the ANOVA are again interpreted as significant. I would recommend to transparently state, that the main effects are not significant but some of the comparisons between time points reach significance. Furthermore, I would sincerely recommend to reinstate the report of the F-values for all main effects.
Author Response
Response to Reviewer 1 Comments
Dear Colleague,
Thank you for the time and effort you dedicated in order to improve the quality of this paper.
Point 1: thank you very much for your revised version of the manuscript. I think the manuscript is quite improved. Nevertheless, I think it is quite problematic that the non-significant main effects of the ANOVA are again interpreted as significant. I would recommend to transparently state, that the main effects are not significant but some of the comparisons between time points reach significance. Furthermore, I would sincerely recommend to reinstate the report of the F-values for all main effects.
Response 1:
We really appreciate this comment. We totally agree that the purpose of the manuscript is to indicate that only some comparisons are significant and overall effects are not (at least for most comparisons), as elapsed time per se was identified to be an independent, negative predictor of sleep disturbances, during follow up. In order to be more clear, we have emphasized the effect of time in the abstract of the manuscript, which now runs: “Elapsed time was found to be negatively and independently associated to global PSQI, PSQI C5-Sleep disturbance, PSQI C7-Daytime dysfunctions….” and we further amended the last sentence of the abstract which now reads: “Conclusions: Several sleep disturbances were observed after hospital discharge for COVID-19 pneumonia during certain time points; However, improvement over time was remarkable in most domains of assessed questionnaires”. Moreover, we have added in the first paragraph of the discussion the following information: “However, apart from the PSQI domain of daytime dysfunction, and the severity of fatigue as assessed by FSS, most sleep disturbances tend to improve with time in our study. To the authors’ knowledge, this is the first longitudinal study to indicate that elapsed time is independently and negatively associated with the presence of sleep disturbances during a six-month follow up, after hospitalization for COVID-19 pneumonia.” Finally, we amended the first sentence of the last paragraph in the section of discussion, which now runs: “To conclude, sleep disturbance and fatigue are often present after hospitalization for COVID-19 pneumonia; However, most, but not all of these disturbances tend to improve with time”.
As far as the second part of the Reviewer’s comment is concerned, we modified the text according to this suggestion, in an effort to present the ANOVA results as clearly as possible. We also reinstated the report of the F-values for all main effects, as proposed.
Should you have any additional remarks or questions, please do not hesitate to contact me.
Kind regards,
Evgenia Kalamara